# Reactive Dicarbonyl Scavenging Effectively Reduces MPO-Mediated Oxidation of HDL and Restores PON1 Activity

**DOI:** 10.3390/nu12071937

**Published:** 2020-06-30

**Authors:** Jiansheng Huang, Patricia G. Yancey, Huan Tao, Mark S. Borja, Loren E. Smith, Valentina Kon, Sean S. Davies, MacRae F. Linton

**Affiliations:** 1Department of Medicine, Division of Cardiovascular Medicine, Atherosclerosis Research Unit, Vanderbilt University Medical Center, Nashville, TN 37232, USA; Jiansheng.huang@vumc.org (J.H.); patricia.g.yancey@vumc.org (P.G.Y.); huan.tao@vumc.org (H.T.); 2Department of Chemistry & Biochemistry, California State University East Bay, Hayward, CA 94542, USA; mark.borja@csueastbay.edu; 3Department of Anesthesiology, Vanderbilt University Medical Center, Nashville, TN 37232, USA; loren.e.smith@vumc.org; 4Department of Pediatrics, Vanderbilt University Medical Center, Nashville, TN 37232, USA; valentina.kon@vumc.org; 5Department of Pharmacology, Vanderbilt University, Nashville, TN 37232, USA; sean.davies@Vanderbilt.Edu

**Keywords:** high-density lipoprotein (HDL), malondialdehyde (MDA), reactive dicarbonyl scavengers, inflammation, cholesterol efflux, macrophages, familial hypercholesterolemia

## Abstract

Atheroprotective functions of high-density lipoproteins (HDL) are related to the activity of HDL-associated enzymes such as paraoxonase 1 (PON1). We examined the impact of inhibition of myeloperoxidase (MPO)-mediated HDL oxidation by PON1 on HDL malondialdehyde (MDA) content and HDL function. In the presence of PON1, crosslinking of apoAI in response to MPO-mediated oxidation of HDL was abolished, and MDA-HDL adduct levels were decreased. PON1 prevented the impaired cholesterol efflux capacity of MPO-oxidized HDL from *Apoe^−/−^* macrophages. Direct modification of HDL with MDA increased apoAI crosslinking and reduced the cholesterol efflux capacity. MDA modification of HDL reduced its anti-inflammatory function compared to native HDL. MDA-HDL also had impaired ability to increase PON1 activity. Importantly, HDL from subjects with familial hypercholesterolemia (FH-HDL) versus controls had increased MDA-apoAI adducts, and PON1 activity was also impaired in FH. Consistently, FH-HDL induced a pro-inflammatory response in *Apoe^−/−^* macrophages and had an impaired ability to promote cholesterol efflux. Interestingly, reactive dicarbonyl scavengers, including 2-hydroxybenzylamine (2-HOBA) and pentyl-pyridoxamine (PPM), effectively abolished MPO-mediated apoAI crosslinking, MDA adduct formation, and improved cholesterol efflux capacity. Treatment of hypercholesterolemic mice with reactive dicarbonyl scavengers reduced MDA-HDL adduct formation and increased HDL cholesterol efflux capacity, supporting the therapeutic potential of reactive carbonyl scavenging for improving HDL function.

## 1. Introduction

Familial hypercholesterolemia (FH) is an autosomal dominant disorder, most commonly due to mutations in the genes for *LDLR*, *ApoB* and *PCSK9*, characterized by remarkably increased levels of low-density lipoprotein cholesterol (LDL-C) and high risk of premature coronary artery disease [1]. The high risk of premature cardiovascular disease (CVD) is attributable to the increased levels of LDL-C and oxidized LDL, with little attention paid to the role of high-density lipoproteins (HDL) function. The relationship between HDL-C levels and risk for atherothrombosis is complex, as there are apparent exceptions to the inverse relationship at both ends of the HDL-C spectrum [2]. Furthermore, recent evidence suggests that HDL particle number may be a better gauge than HDL-C levels for CVD protection. Recent studies have shown that HDL cholesterol efflux capacity is inversely associated with carotid and coronary atherosclerosis and atherosclerotic cardiovascular events independent of HDL-C levels [3,4,5,6].

Myeloperoxidase (MPO) is a heme-containing enzyme released from azurophilic granules of polymorphonuclear neutrophils and monocytes into the extracellular fluid in the setting of inflammation. The MPO-derived hypochlorous acid (HOCl) oxidizes HDL in human atherosclerotic lesions and reduces ABCA1-dependent cholesterol efflux by site-specific chlorination of apolipoprotein A-I (apoAI) tyrosine residues [7,8,9,10]. In addition, modification of HDL by MPO generates a pro-inflammatory particle [11,12]. These findings suggest that MPO peroxidase activity promotes the formation of dysfunctional HDL in human atherosclerosis. Besides direct oxidation of apoAI, MPO derived oxidants also modify lipids generating highly reactive dicarbonyls such as malondialdehyde (MDA) and isolevuglandins (IsoLGs), which covalently bind apoAI and reduce cholesterol efflux (Figure 1) [13,14]. Therefore, we examined the ability of the dicarbonyl scavengers, 2-hydroxylbenzylamine (2-HOBA) and pentyl-pyridoxamine (PPM), to modulate HDL function by neutralizing reactive dicarbonyls (Figure 1).

The chronic inflammatory atherosclerotic process has been associated with reduced activity of the HDL-associated anti-oxidative enzyme, paraoxonase 1 (PON1) [15]. PON1 is associated with increased plasma HDL levels [16]. While the in vivo substrates of PON1 remain to be identified, PON1 activity (Figure 1) prevents the accumulation of lipid hydroperoxides in HDL and LDL [17]. PON1 also inhibits N-homocysteinylation of LDL-associated proteins by hydrolyzing the highly reactive pro-oxidant homocysteine thiolactone [18]. In addition, PON1 activity neutralizes bioactive oxidized 1-palmitoyl-2-arachidonoyl phosphatidylcholine [19]. Recent studies found that apoAI forms a functional ternary complex with PON1 and MPO [20]. However, the impact of inhibition of MPO-mediated HDL oxidation by PON1 on HDL function is still unknown. We examined the impact of HDL-associated PON1 in preventing MPO-mediated HDL MDA modification and dysfunction, and the potential role of reduced PON1 activity in promoting increased MDA modification and impaired function of HDL in FH patients. Here, our findings demonstrate that PON1 inhibition of MPO activity prevents MDA-HDL adduct formation and crosslinking of apoAI and preserves the cholesterol efflux capacity of HDL. In addition, direct modification of native HDL with MDA increases apoAI crosslinking and impairs the ability of HDL to enhance PON1 activity, reduces inflammation, and mediates cholesterol efflux. Interestingly, subjects with FH have decreased PON1 activity, and FH-HDL contains cross-linked apoAI and MDA-apoAI adducts that likely contribute to the impaired cholesterol efflux and anti-inflammatory functions of FH-HDL. Importantly, administration of reactive dicarbonyl scavengers to hypercholesterolemic mice increased PON1 activity and HDL cholesterol efflux capacity, demonstrating the therapeutic potential of reactive dicarbonyl scavengers in preserving HDL atheroprotective functions.

## 2. Materials and Methods

*Reagents*—Human neutrophil myeloperoxidase (MPO) was obtained from Lee Biosolutions Inc. (St. Louis, MO, USA). Human paraoxonase 1 (PON1) recombinant protein was purchased from Thermo Fisher Scientific Inc. (Waltham, MA, USA). Human apolipoprotein A-I (apoAI), and dimethyl sulfoxide (DMSO) were obtained from Alfa Aesar (Ward Hill, MA, USA). Fresh HDL was purified from FH patients and normal subjects following the HDL purification kit’s instructions (Cell Biolabs Inc., California, USA). Maloncarbonyl bis-(dimethylacetal), taurine and H_2_O_2_ were purchased from Sigma-Aldrich (St. Louis, MO). PON1 ELISA kit and EnzCheck PON1 activity kit were purchased from R&D systems Inc. (Minneapolis, MN, USA) and Invitrogen (Carlsbad, CA, USA), respectively. The MDA-HDL enzyme-linked immunosorbent assay (ELISA) kit was purchased from Cell Biolabs Inc. (San Diego, California). The chemiluminescent Western Lightning ultra-reagent was obtained from PerkinElmer Inc. (Waltham, MA, USA). L-012 was obtained from Wako Chemicals (Richmond, VA, USA). The primary antibody to apoAI was purchased from Meridian Life Science Inc. (Memphis, TN, USA), and the secondary antibody horseradish peroxidase (HRP)-conjugated anti-goat IgG from rabbit was purchased from Sigma-Aldrich (St. Louis, MO, USA).

Working solutions of H_2_O_2_ were made fresh daily by diluting 30% H_2_O_2_ (BDH Chemicals, London, UK) according to the extinction coefficient for H_2_O_2_ at 240 nm, 39.4 M^−1^cm^−1^ [21]. Buffers were used in L-012 luminescence assay, including PS6.5 buffer (130 mM NaCl, 6.28 mM Na_2_HPO_4_, 18.7 mM NaH_2_PO_4_), and MAPS imaging solution (buffer Cit6 with 20 mM NaBr, 200 mM (NH_4_)_2_SO_4_, 100 p.p.m. *v/v* Tween20, 50 μM L-012 and 100 μM (H_2_O_2_), 1 ppm = 0.0001%)) [22,23].

*Mice—Apoe^−/−^ and Ldlr^−/−^* mice on a C57BL/6 background were purchased from Jackson Laboratories (Bar Harbor, ME). Mice were maintained in microisolator cages with ad libitum access to a rodent chow diet containing 4.55% fat (PMI 5010, St. Louis, Mo) and water. For in vivo studies, after pretreatment of mice on a chow diet with 2-HOBA for 2 weeks, 7-week-old female *Ldlr^−/−^* mice were placed on a western diet and treated with 1 g/L of 2-HOBA for 16 weeks. Serum samples were collected at sacrifice. The mouse protocols were approved by the Institutional Animal Care and Use committee of Vanderbilt University. Experimental procedures and animal care were performed according to the regulations of Vanderbilt University’s Institutional Animal Care and Usage Committee.

*Reactive Dicarbonyl Scavengers*—We used 2-HOBA and PPM to examine the effects of reactive dicarbonyl scavenging on HDL function in vitro and in vivo. In some experiments, taurine was used as a scavenger. PPM, 2-HOBA, and taurine were easily solubilized in water and prepared as a 100 mM stock and stored as small aliquots in −80 °C until use. Fresh working solutions were prepared before each assay and diluted in water to appropriate concentrations.

*Human Blood Collection, ApoB-Depleted Serum Preparation, and Lipoprotein Isolation*—The study was approved by the Vanderbilt University Institutional Review Board (IRB), and all participants gave their written informed consent. The human blood from FH patients and healthy controls were obtained using an IRB approved protocol. ApoB-depleted serum was prepared as described [24]. Briefly, to remove apoB-containing lipoproteins from the serum of fasted subjects, 400 μL of PEG 8000 (Sigma-Aldrich, St. Louis, MO] (20% in 200 mM glycine) was added to 1 mL serum, incubated for 15 min at room temperature, and then centrifuged at 1900 g for 20 min. The supernatant containing the apoB-depleted serum was collected and used. HDL and LDL were prepared from serum by Lipoprotein Purification Kits (Cell BioLabs, Inc., San Diego, CA, USA).

*Peritoneal Macrophages*—Peritoneal macrophages were isolated from 3 to 4-month-old *Apoe^−/−^* mice (Jackson Laboratories, Sacramento, CA). Four days after injection of 10% of thioglycolate, cells were obtained by peritoneal lavage with ice-cold Ca^2+^ and Mg^2+^ free PBS. Peritoneal cells were centrifuged and re-suspended in DMEM supplemented with 10% heat-inactivated fetal bovine serum (FBS) and 100 units/mL penicillin/streptomycin. Cells were plated onto 24-well culture plates (1 × 10^6^ cells/well) and allowed to adhere for 2 h. Non-adherent cells were removed by washing two times with DPBS, and adherent macrophages were used for experiments.

*Modification of apoAI and HDL with MDA*—MDA was prepared immediately before use by rapid acid hydrolysis of maloncarbonyl bis-(dimethylacetal) as described [13]. Briefly, 20 μL of 1 M HCl was added to 200 μL of maloncarbonyl bis-(dimethylacetal), and the mixture was incubated for 45 min at room temperature. The MDA concentration was determined by absorbance at 245 nm, using the coefficient factor 13,700 M^−1^ cm^−1^. ApoAI (1 mg protein / mL) or HDL (10mg of protein /mL) and increasing doses of MDA (0, 0.125 mM, 0.25 mM, 0.5 mM, 1 mM) were incubated at 37 °C for 24 h in 50 mM sodium phosphate buffer (pH7.4) containing diethylenetriamine pentaacetate(DTPA) 100 μM. Reactions were initiated by adding MDA and stopped by dialysis of samples against phosphate buffer saline (PBS) at 4 °C.

*Modification of HDL with MPO*—The MPO-mediated HDL modification reactions were performed in 50 mM phosphate buffer (pH 7.0) containing 100 μM DTPA, 1 mg/mL protein of HDL, 70 nM purified human MPO (A430/A280 ratio of 0.79), 80 uM H_2_O_2_, and 100 mM chloride [25]. For the PON1 inhibition reaction, 1 U/mL PON1 was incubated with 1 mg/mL HDL for 10 min and then 70 nM MPO, 80 uM H_2_O_2_, and 100 mM NaCl were added to initiate the reaction. The reactions were carried out at 37 °C for 1 h. In all reactions, the concentration of the key reactants was verified spectrophotometrically using molar extinction coefficients of 170 cm^−1^ mM^−1^ at 430 nm for MPO and 39.4 cm^−1^ mM^−1^ at 240 nm for H_2_O_2_.

*MDA-apoAI and MDA-HDL Adduct ELISA*—The ELISA assays for MDA adduct levels were performed essentially following the manufacturer’s instructions (Cell Biolabs, Inc.). Briefly, 96-well coated plates were blocked by adding 100 μL of blocking buffer (1% BSA, 0.05% NaN3, in PBS, pH 7.2–7.4) and incubated at room temperature for 2 h. HDL was isolated using a dual precipitation kit (Cell Biolabs, Inc., San Diego, CA, USA). The diluted HDL samples and standard curve MDA-HDL samples (5 ng/mL) were then added to wells and incubated for 2 h at room temperature with shaking at 600–800 rpm. After washing, 100 μL of detection antibody was added to the wells and incubated for 2 h. Then HRP–streptavidin was added and incubated for 30 min at room temperature. A total of 100 μL of substrate was added and incubated at room temperature for 5 min. After stopping the reaction, the optical density (OD) values were determined at 450 nm.

*PON1 ELISA*—The PON1 levels were measured using the ELISA assay (R&D systems, Inc.) following the manufacturer’s instructions with a few modifications. Briefly, unknown serum samples (diluted 1:500) and the PON1 standards were added to wells coated with the PON1 capture antibody and incubated for 2 h. Each well was then rinsed 5 times with 200 μL of washing buffer, and 100 μL of blocking solution was added to each well. After incubation for 2 h at room temperature, the biotinylated goat anti-human PON1 antibody (diluted 1:1000) was added to detect the captured PON1 from serum and incubated for 2 h. The wells were then washed 5 times, incubated for 1 h with streptavidin-HRP conjugate, then incubated with substrate solution for 20 min at room temperature. After stopping the reaction, the OD was measured at 450 nm wavelength.

*Analysis of PON1 Activity*—The PON1 activity assay was performed using the EnzCheck PON1 activity kit (Thermo Fisher Scientific), which measures the organophosphatase activity of PON1. PON1 standards and diluted human serum samples were prepared in Tris-saline calcium (TSC) buffer (20 mM Tris-HCl, 150 mM NaCl and 2 mM CaCl_2_, pH 8.0). The samples and standards were then incubated for 30 min with TSC buffer containing 100µM substrate. Fluorescence intensity was then measured using Biotek’s Synergy MX Microplate reader (Winooski, VT) with Gen 5 software at excitation wavelength of 368 nm and an emission wavelength of 460 nm after 30 min of incubation at 37 °C.

*MPO Peroxidase Activity Assay*—The MPO luminescence assay was performed as described [26]. To evaluate the inhibitory effect of PON1 on MPO activity, a luminescence assay was developed with L-012 as the peroxidase substrate. L-012 oxidation was monitored as a function of luminescence intensity over time using Biotek’s Synergy MX Microplate reader (Winooski, VT) with Gen 5 software. PON1 (0, 0.12, 0.24, 0.48, 0.94, 1.87, 3.75, 7.5 uM) was incubated with MPO for 30 min at room temperature, the samples were loaded into microtiter plate wells, and then freshly prepared MPO activity on a polymer surface (MAPS) imaging solution containing L-012 and H_2_O_2_ was added. Each well received 75 μL imaging solution and the plate was briefly shaken up to 1000 rpm and imaged immediately. Data were quantitated at the 5-min time-point.

*HDL Cholesterol Efflux Capacity Assay*—Cholesterol efflux was measured as previously described [27,28]. Briefly, *Apoe^−/−^* macrophages were isolated and incubated for 48 h with Dulbecco’s Modified - Eagle’s Medium (DMEM) containing acetylated LDL (40 μg/mL) and 1.0 μCi/mL ^3^H-cholesterol (PerkinElmer, Boston, MA). The cells were then washed and incubated for 24 h in DMEM supplemented with 4-(2-hydroxyethyl)-1-piperazineethanesulfonic acid (HEPES) in the presence of HDL or MDA-HDL. After filtering aliquots of media through 0.45µm multiscreen filtration plates to remove floating cells, the [^3^H]cholesterol was measured by liquid scintillation counting. Cellular [^3^H]cholesterol was extracted by incubating the monolayers overnight in isopropanol. Cellular cholesterol content and proteins were measured as described [13,14].

*Measurement of HDL-apoAI Exchange (HAE)*—HDL at a concentration of 1 mg/mL (total protein) was mixed with 3 mg/mL spin-labeled, lipid-free apoAI in a 3:1 ratio [29,30]. Samples were drawn into borosilicate capillary tubes (VWR) and incubated for 15 min at 37 °C. Electron paramagnetic resonance (EPR) measurements were performed in a Bruker EMX Nano spectrometer outfitted with a temperature controller set to 37 °C. The peak amplitude of the nitroxide signal (3462–3470 Gauss) was compared with the peak amplitude of a proprietary internal standard (3507–3515 Gauss) provided by Bruker. The internal standard is contained within the instrument and does not contact the sample. Because the *y* axis of the EPR spectrometer is measured in arbitrary units, measuring the sample against a fixed internal standard facilitates normalization of the response. The HDL-apoAI exchange (HAE) activity represents the sample/internal standard signal ratio at 37 °C. The maximal percentage of HAE activity was calculated by comparing HAE activity with a standard curve ranging in the degree of spin-labeled, lipid-associated apoA-I signal. Experiments were repeated in duplicate and averaged.

*Measurement of free lysine using o-phthalaldehyde (OPA)*—OPA is a primary amine-reactive fluorescent detection reagent that is used to detect free lysine in HDL [14,31,32]. The procedure was performed according to the manufacturer’s instructions (Thermo Scientific) using HDL modified by lipid aldehydes as described above and adapted to 96-well plates. The % lysine adduction was calculated as fluorescence of modified HDL/unmodified HDL.

*Western Blot for apoAI Crosslinking*—Protein samples were incubated with reducing reagent β-mercaptoethanol and polyacrylamide gel electrophoresis (SDS-PAGE) sample loading buffer for 10 min at 55 °C and then resolved by NuPAGE Bis-Tris electrophoresis. The gels were transferred onto polyvinylidene difluoride (PVDF) membranes (Amersham Bioscience) at 150 V for 1.5 h. Blots were blocked with 5% milk at room temperature for 2 h and probed with primary antibodies specific to human apoAI from goat overnight at 4 °C. The secondary antibodies conjugated with HRP were incubated with the membranes for 2 h at room temperature. Protein bands were visualized with Enhanced chemiluminescence (ECL) Western blotting detection reagents (GE Healthcare).

*RNA isolation and Quantitative RT-PCR—Apoe^−/−^* macrophages were incubated for 4 h with 25 ng/mL lipopolysaccharide(LPS) in the absence or presence of 50 μg/mL of HDL or MDA-HDL (molar ratios of MDA to HDL are 0/1, 5/1, 10/1, 20/1, 40/1). Total RNA was extracted from peritoneal macrophages using the RNeasy mini kit (Qiagen, Valencia, CA, USA) and first-strand cDNA was synthesized from the total RNA (250 ng) using a reverse transcription reagent (Applied Biosystems, Foster City, CA). Quantitative PCR was performed with a Perkin–Elmer 7900 PCR machine, TaqMan PCR master mix and 6-carboxyfluorescein (FAM)-labeled TaqMan probes (Assays-on-Demand, Applied Biosystems) for IL-6, IL-1β, and glyceraldehyde-3-phosphate dehydrogenase (GAPDH). Samples were run in 20 µL reactions using an ABI 7800 (Applied Biosystems). Samples were incubated at 95 °C for 15 min, followed by 40 cycles at 95 °C for 10 s, 56 °C for 20 s, and 72 °C for 30 s. Expression data were normalized to GAPDH levels.

*Statistical Analysis*—Results are provided as mean ± the standard error of the mean (SEM). Data were examined by the Kolmogorov–Smirnov test for normality, then the differences between mean values were determined by Student’s t test or one-way ANOVA coupled with the Bonferroni–Dunn post hoc test using Prism 6. *p* < 0.05 was considered to be statistically significant.

## 3. Results

### 3.1. Effects of PON1 on MPO Activity, apoAI Crosslinking, MDA-HDL Adducts, and HDL Cholesterol Efflux Capacity

The MPO peroxidase activity was maximally inhibited by increasing doses (IC50 = 40 nM) of PON1 (Figure 2A), which is consistent with previous studies [20]. The addition of MPO and its substrates to HDL increased apoAI crosslinking (Figure 2B) [8]. However, this MPO-mediated crosslinking of apoAI in HDL was almost abolished in the presence of 100 nM PON1 (Figure 2B). MPO also increased the production of MDA-HDL adducts seven-fold (*p* < 0.05) compared to the MDA content of HDL not treated with MPO (Figure 2C). Importantly, the addition of PON1 reduced MPO-mediated MDA-HDL adduct production by 86% (*p* < 0.05) (Figure 2C). Consistent with the increased apoAI crosslinking and MDA-HDL adducts, MPO-mediated oxidation of HDL reduced its cholesterol efflux capacity by 56% (Figure 2D). The inhibition of MPO activity by PON1 prevented the impairment in the cholesterol efflux capacity of HDL (Figure 2D). Taken together, these results suggest that PON1 preserves the HDL cholesterol efflux function by inhibiting MPO-mediated apoAI crosslinking and MDA-HDL adduct formation.

### 3.2. Effects of MDA Modification of apoAI or HDL on apoAI Crosslinking, Cholesterol Efflux Capacity, and HDL-apoAI Exchangeability

As our data demonstrate that PON1 inhibits MPO-mediated apoAI crosslinking and MDA-HDL adduct production, we next examined the effects of direct modification of apoAI or HDL with MDA on apoAI crosslinking and cholesterol efflux. ApoAI (molar ratios of MDA to apoAI: 25/1, 50/1, 100/1) and HDL (molar ratios of MDA to apoAI: 5/1, 10/1, 20/1, 40/1) were directly modified with increasing doses of MDA. Direct modification of apoAI (Appendix A) and HDL (Appendix A) with MDA caused crosslinking of apoAI in a dose dependent manner. In addition, direct MDA modification of apoAI (Appendix A) and HDL (Appendix A) led to similar MDA adduct content, as was observed with MPO-mediated oxidation of HDL (Figure 2C). Modification with MDA reduced the cholesterol efflux capacity of apoAI (Appendix A) by 21.8% (*p* < 0.05), 22% (*p* < 0.05), 55% (*p* < 0.001), for apoAI modified with increasing MDA concentrations (molar ratio of MDA to apoAI of 25/1, 50/1, 100/1, respectively), which is consistent with other studies [13]. In addition, modification with MDA reduced the cholesterol efflux capacity of HDL (Appendix A) by ~ 37% (*p* < 0.05), 42% (*p* < 0.05), 62.8% (*p* < 0.01), and 80% (*p* < 0.001), respectively, for 1 mg/mL HDL modified with MDA (molar ratios of MDA/HDL equivalent to apoAI: 0/1, 1/1 5/1, 20/1, 40/1). We determined the ability of MDA to modify lysine residues on HDL using o-phthalaldehyde (OPA) to detect available lysines. OPA also detects the headgroups of phosphatidylethanolamines (PEs), although these are in much lower abundance than lysyl residues. Modification with MDA (40 eq) reduced the available lysines of HDL by approximately 40% (Appendix A). As some of the cholesterol efflux capacity of HDL is from dissociated apoAI interacting with ABCA1 [33], we also performed electron paramagnetic resonance (EPR) to measure the ability of HDL-apoAI to exchange [29]. Interestingly, modification with MDA (molar ratios of MDA/HDL equivalent to apoAI: 0/1, 1/1 5/1, 20/1, 40/1) reduced the HDL-apoAI exchangeability by 32%, 69%, 64%, 67%, and 71%, respectively (Appendix A). Taken together, these data indicate that MDA modification of HDL increases HDL-apoAI crosslinking, resulting in impaired cholesterol efflux and reduced HDL-apoAI exchangeability.

### 3.3. Effects of Reactive Dicarbonyl Scavenging on HDL-apoAI Crosslinking and Cholesterol Efflux Capacity during MDA Modification and MPO-Mediated Oxidation

We first determined whether the reactive dicarbonyl scavenger, taurine, could prevent the crosslinking of apoAI in HDL directly modified by MDA. The treatment of HDL with 5 mM taurine before the addition of MDA (250 μM MDA) resulted in reduction in apoAI crosslinking (Figure 3A). Similar results were seen when apoAI was pretreated with taurine and modified with MDA (data not shown). Consistent with the reactive dicarbonyl scavenging reducing HDL-apoAI crosslinking, the cholesterol efflux capacity of taurine-treated HDL was preserved (Figure 3B). Importantly, reactive dicarbonyl scavenging with 5 mM taurine, 500 μM 2-HOBA, or 500 μM PPM markedly reduced MPO-mediated HDL apoAI crosslinking (Figure 3C–E). In addition, 500 μM 2-HOBA prevented the MPO-mediated HDL cholesterol efflux dysfunction (Figure 3F).

### 3.4. Effects of HDL MDA Modification and Dicarbonyl Scavenging on PON1 Activity

It has been shown that PON1 interacts with apoAI of HDL resulting in increased enzymatic activity [34]. Native HDL increased the activity of recombinant PON1 by 56% (Figure 4A), whereas MDA modification of HDL reduced its ability to activate PON1, with the highest dose of MDA only increasing activity by 12% (molar ratio of MDA to HDL: 40/1). As administration of an atherogenic diet to *Ldlr^−/−^* mice increases apoAI oxidation and crosslinking and reduces PON1 activity [35], we examined whether scavenging reactive dicarbonyls improves PON1 activity and HDL cholesterol efflux capacity in *Ldlr^−/−^* mice consuming a western diet. The treatment of *Ldlr^−/−^* mice with 1g/L 2-HOBA improved the PON1 activity by 18.5% (Figure 4B). In addition, the treatment of the *Ldlr^−/−^* mice with the reactive dicarbonyl scavenger, PPM, increased the cholesterol efflux capacity of their HDL by 37.5% (Figure 4C). Consistent with the improved PON1 activity and HDL cholesterol efflux function, the HDL MDA content was reduced by 61% (Figure 4D). In addition, the MDA content of LDL decreased by 57% (Figure 4E).

### 3.5. FH Patients Have Decreased PON1 Activity, Increased MDA-apoAI Adducts, Defective HDL Cholesterol Efflux Capacity and Impaired HDL Anti-Inflammatory Function

As hypercholesterolemia impacts PON1 activity in mice, we next examined the PON1 levels and activity in FH patients versus control subjects. After precipitation of the apoB containing lipoproteins from serum with polyethylene glycol(PEG), PON1 mass and activity were measured in the HDL containing fraction. Compared to control subjects, PON1 levels and activity were decreased by 23% (*p* < 0.05) and 41.7% (*p* < 0.05) in the FH-HDL fraction, respectively (Figure 5A,B). The PON1 activity and levels were not significantly different in the FH-HDL fractions (Figure 5A,B) of serum collected pre- versus post-LDL apheresis (LA). Normalization of the PON1 activity to PON1 mass revealed a 24% decrease in specific activity, indicating that PON1 activity is impaired in FH-HDL (Figure 5C). In addition, the levels of MDA-apoAI adducts were increased nearly three-fold in FH versus control HDL (Figure 5D). In agreement with the increased MDA adducts, the cholesterol efflux capacity of FH-HDL was markedly reduced compared to control HDL (Figure 5E).

Because reduced PON1 inhibition of MPO-mediated HDL oxidation produces pro-inflammatory particles [11], we next examined the effects of direct MDA modification of HDL on the response to LPS in *Apoe^−/−^* macrophages. While control HDL inhibited the LPS-induced expression of IL-1β and IL-6 by 60% (*p* < 0.05) and 40% (*p* < 0.05) (Figure 6A,B), respectively, MDA-HDL (molar ratio of MDA to HDL of 40/1) enhanced the expression of IL-1β and IL-6 3-fold (*p* < 0.05) and 1.8-fold (*p* < 0.05), respectively. In addition, MDA reduced the anti-inflammatory function of HDL in a dose-dependent manner (Figure 6A,B). We next examined the effectiveness of FH-HDL versus control HDL in preventing the inflammatory response to LPS in *Apoe^−/−^* macrophages. Unlike control HDL, which reduced the expression of IL-1β and TNF-α in response to LPS, FH-HDL, from subjects prior to LDL apheresis (LA), increased the mRNA levels of IL-1β and TNF-α 400-fold and 10-fold when compared to incubation with LPS alone, respectively (Figure 6C,D), which indicates that FH-HDL is extremely pro-inflammatory. Similarly, FH-HDL post-LA induced a pro-inflammatory response to LPS (Figure 6C,D). Thus, it is likely that MDA-HDL adducts contribute to the inflammatory nature of both MPO-oxidized HDL and FH-HDL. More importantly, recent studies have shown that oxidized LDL (ox-LDL) induces generation of reactive oxygen species in cells, in particular, H_2_O_2_, which is necessary for the inflammatory response to oxidized LDL [36,37]. Therefore, we examined whether a reactive dicarbonyl scavenger could block the pro-inflammatory cytokine response of peritoneal macrophages to ox-LDL, as reflected by reduced mRNA levels of IL1β and IL-6, which are relevant to the development of atherosclerosis. In the presence of 2-HOBA, macrophages were incubated with 50 µg/mL oxLDL for 24 h, and then the effects on macrophage expression of IL-1β and IL-6 in response to oxLDL were measured. Interestingly, 2-HOBA significantly reduced the ox-LDL-induced mRNA expression levels of pro-inflammatory cytokines IL-1β and IL-6 in *Apoe^−/−^* macrophages (Figure 7).

## 4. Discussion

Oxidative modifications of HDL proteins and lipids result in compromised function, and evidence has mounted that HDL dysfunction is atherogenic. The current study shows that HDL-associated PON1 is effective at preventing MPO-mediated HDL modification and apoAI crosslinking, which preserves HDL anti-inflammatory and cholesterol efflux functions. In addition, we show that MDA modification of HDL results in reduced ability to activate PON1, which may be relevant to FH patients who have increased plasma MDA-HDL adducts that likely contribute to their decreased PON1 activity and HDL dysfunction. In vitro inhibition of MPO-generated reactive dicarbonyls with the scavengers, PPM and 2-HOBA, prevented HDL apoAI crosslinking and cholesterol efflux dysfunction. Importantly, in vivo scavenging of reactive dicarbonyls improved PON1 activity and HDL cholesterol efflux function, supporting this as a potential therapeutic strategy for reducing atherosclerosis.

MPO mediates the chlorination, nitration, and direct oxidation of proteins as well as initiating lipid peroxidation that causes the generation of highly reactive dicarbonyls (Figure 1). Plasma MPO levels predict risk of clinical events in subjects with CVD, and human atherosclerotic lesions are markedly enriched in MPO activity [38,39,40]. MPO interacts directly with apoAI, and this interaction is enhanced with MPO-oxidized apoAI [41], causing accumulation of apoAI that is extensively crosslinked and dysfunctional within atherosclerotic lesions. Recent studies showed that PON1 forms a ternary complex with apoAI and MPO, and that PON1 interaction with apoAI decreases MPO activity [20]. In addition, interaction of PON1 with apoAI activates the enzyme and enhances its ability to hydrolyze oxidized phospholipids. The current studies show that the addition of PON1 to HDL and MPO essentially abolishes MPO-mediated apoAI crosslinking and MDA adduct content, thereby effectively maintaining HDL cholesterol efflux capacity (Figure 2). Interestingly, we show that MDA modification of HDL reduced the ability of HDL to activate PON1 (Figure 4A), raising the possibility that MDA adduct formation interferes with apoAI activation of PON1. In this regard, a prior study showed that MDA targets the K_206_ and K_208_ residues of lipid-free apoAI to form lysine-MDA-lysine adducts [13], and studies by Huang and colleagues [20] demonstrated that the peptide region, S_201_TLSEKAK_208_, is critical for PON1 interaction with apoAI. Indeed, mutations in this region (S_201_TLSEKAK_208_ to S_201_ALAAEAE_208_) are designed to decrease ionic interactions and markedly reduce PON1 interaction and activation, resulting in an impaired ability to decrease MPO activity [20]. The MDA-mediated modification of HDL is likely due primarily to lysine-mediated crosslinks, whereas the MPO-mediated modification of HDL is probably also due to the MPO-derived HOCl (e.g., di-tyrosine). A recent study found that MPO-derived products, such as protein-bound oxidized tyrosine moieties 3-chlorotyrosine, 3-nitrotyrosine, and o,o′-dityrosine, are very important for in chronic kidney disease (CKD)-accelerated atherosclerosis [42]. Taken together, it is likely that modification of apoAI with lysine-MDA-lysine adducts impairs PON1 interaction with HDL. Consistent with this possibility, the treatment of *Ldlr^−/−^* mice with reactive dicarbonyl scavengers significantly increased their PON1 activity and decreased MDA-HDL adducts (Figure 4).

The current studies demonstrate that MPO-mediated oxidation of HDL increases the levels of MDA-HDL adducts (Figure 2). Direct MDA modification of lipid-free apoAI caused intermolecular crosslinking (Appendix A), and, consistent with other studies, MDA-apoAI adduct formation decreased cholesterol efflux in a MDA dose-dependent manner [13]. Similar to MDA-apoAI, MDA modification of HDL caused extensive apoAI crosslinking. In addition, MDA modification of HDL likely generated apoAI/apoAII heterodimers, as shown by the presence of a 38kDa band that was absent with MDA-apoAI (Appendix A). Similar to MDA-apoAI, MDA modification of HDL impaired its cholesterol efflux capacity (Appendix A). MDA-HDL also had a reduced ability to prevent the macrophage inflammatory response to LPS (Figure 6). MDA-HDL had decreased apoAI exchangeability, which likely contributes to the impaired cholesterol efflux and anti-inflammatory functions. Studies have shown that a significant portion of cholesterol efflux to HDL occurs via ABCA1, which releases cholesterol and phospholipids to ApoAI, some of which has dissociated from HDL particles [43,44]. MPO is crucial for the chlorination, nitration, and direct oxidation of proteins. Chlorination of apoA-I by the MPO pathway has been found to affect its ability to remove excess cellular cholesterol through the ABCA1-mediated pathway [45]. It is also known that chlorination blocks the binding of apoA-I to ABCA1 [25]. Furthermore, exchangeable apoAI can reduce the inflammatory response to LPS via ABCA1-mediated activation of STAT3 [46]. Consistent with this concept, HDL function via ABCA1 is correlated with HDL apoAI exchangeability [29]. Studies have also shown that decreased HDL apoAI exchangeability is associated with the presence of atherosclerosis in humans, raising the possibility that MPO-mediated MDA modification of HDL impacts atherosclerosis development [29].

Our studies show that FH-HDL versus control HDL contained three-fold more MDA-lysine adducts. Compared to control HDL, FH-HDL had impaired cholesterol efflux and anti-inflammatory function (Figure 6). In addition, FH patients had decreased PON1 activity compared to controls (Figure 5). The increased MDA content of FH-HDL probably contributes to its dysfunction, as direct MDA modification of control HDL impaired PON1 activity, cholesterol efflux potential, and anti-inflammatory function. However, it is likely that other factors contribute to the dysfunction of FH-HDL, especially given the extremely proinflammatory effects of the particles. A number of apoAI oxidative modifications have been shown to decrease the anti-inflammatory function of HDL, including chlorinated tyrosine and oxidized tryptophan 72 [12,25,31,47]. Furthermore, our recent studies showed that FH versus control HDL contain increased isolevuglandins, which are highly reactive γ-ketoaldehydes [14], and direct modification of HDL with isolevuglandins caused a proinflammatory response to LPS. In addition to oxidation products, other factors such as LCAT, CETP, microRNAs, apoL1, and sphingosine-1-phosphate could impact the anti-inflammatory function of FH-HDL [48,49,50].

Recent studies have demonstrated that HDL function is an independent predictor of coronary artery disease (CAD) [3,4,5,6]. We found that lowering of LDL-C with LDL apheresis has little acute impact on PON1 levels, PON1 activity, or the function of FH-HDL. These findings highlight that in addition to cholesterol lowering drugs, therapeutic treatments that preserve HDL function are needed to further reduce CAD risk. The current studies show that the reactive dicarbonyl scavenger, 2-HOBA, eliminated MPO-mediated HDL apoAI crosslinking, MDA adducts, and cholesterol efflux dysfunction (Figure 3). In addition, the oxidation of HDL may also be alternatively mediated by MPO-derived hypochlorous acid (HOCl). Both 2-HOBA and PPM are electron rich molecules, which may readily react with HOCl [51]. Importantly, in vivo administration of reactive dicarbonyl scavengers to *Ldlr^−/−^* mice consuming a western diet, which increases oxidative stress, enhanced PON1 activity and HDL cholesterol efflux capacity (Figure 4). The results observed in vivo may also be due to the scavenging effect of 2-HOBA on MPO-derived HOCl. Consistent with the improved HDL function, in vivo treatment with reactive dicarbonyl scavengers also reduced the MDA-HDL adducts. Taken together, our studies suggest that reactive dicarbonyl scavengers have therapeutic potential in preventing HDL dysfunction. In addition to preventing MDA-HDL adduct formation, in vivo treatment of *Ldlr^−/−^* mice with PPM markedly reduced MDA-LDL adduct formation (Figure 4). Thus, in vivo reactive dicarbonyl scavenging may also be atheroprotective by preventing reactive aldehyde modification of LDL, which causes enhanced uptake by macrophages and accelerated foam cell formation [52,53].

## 5. Conclusions

In summary, HDL-associated PON1 prevents MPO-mediated HDL modification and apoAI crosslinking, which protects HDL atheroprotective functions. The increased MDA-HDL adducts in FH patients likely contribute to their decreased PON1 activity and HDL dysfunction. The inhibition of MPO generated reactive dicarbonyls, while dicarbonyl scavengers reduced HDL apoAI crosslinking and dysfunction. Finally, in vivo scavenging of reactive dicarbonyls is a novel therapeutic strategy for preserving PON1 activity and maintaining HDL function, which may provide an approach for reducing the residual risk of atherosclerotic cardiovascular disease beyond LDL lowering.

## Figures and Tables

**Figure 1 nutrients-12-01937-f001:**
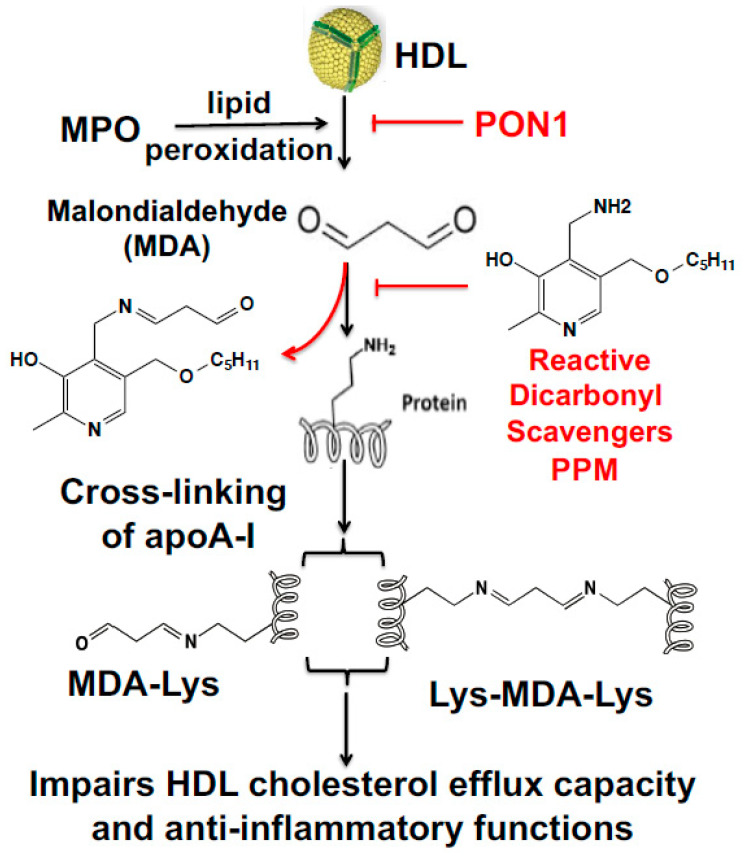
Mechanism of malondialdehyde (MDA) formation from peroxidation of high-density lipoproteins (HDL), its subsequent crosslinking of proteins and effects of reactive dicarbonyl scavenging on HDL cholesterol efflux capacity and anti-inflammatory functions.

**Figure 2 nutrients-12-01937-f002:**
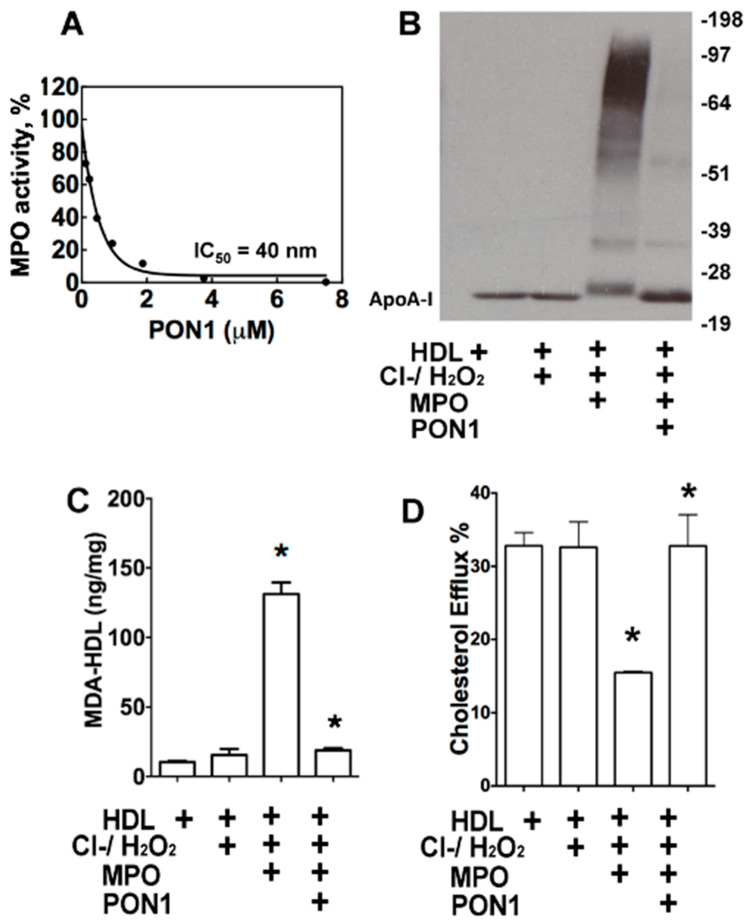
PON1 inhibits MPO-mediated HDL apoAI crosslinking, decreases MDA-HDL adduct levels and preserves HDL function. The MPO-mediated HDL modification reactions were performed in 50 mM phosphate buffer (pH 7.0) containing 100 μM DTPA, 1 mg/mL protein of HDL, 70 nM purified human MPO and 100 mM chloride. For the PON1 inhibition reaction, 1 U/mL PON1 was incubated with 1 mg/mL HDL for 10 min and then 70 nM MPO, 80 uM H_2_O_2_, and 100 mM NaCl were added to initiate the reaction. The reactions were carried out at 37 °C for 1 h. MPO peroxidase activity is inhibited by PON1 as determined by an L-012 biochemical luminescence assay (**A**) and PON1 attenuates the crosslinking of apoAI in HDL oxidized by MPO (**B**). PON1 decreases the levels of MDA-HDL adducts (**C**) and attenuates HDL cholesterol efflux in MPO-mediated HDL oxidation (**D**). (**B**–**D**) Reactions were HDL alone, HDL with 80 µM H_2_O_2_, HDL with 100 mM NaCl, 80 µM H_2_O_2_, and MPO, and HDL with PON1, 100 mM NaCl, 80 µM H_2_O_2_, and MPO. Graphs represent data (mean ± SEM) of three experiments; * *p* < 0.05 is indicated by comparing with HDL group or with MPO-mediated HDL oxidation group (**D**) using one way ANOVA.

**Figure 3 nutrients-12-01937-f003:**
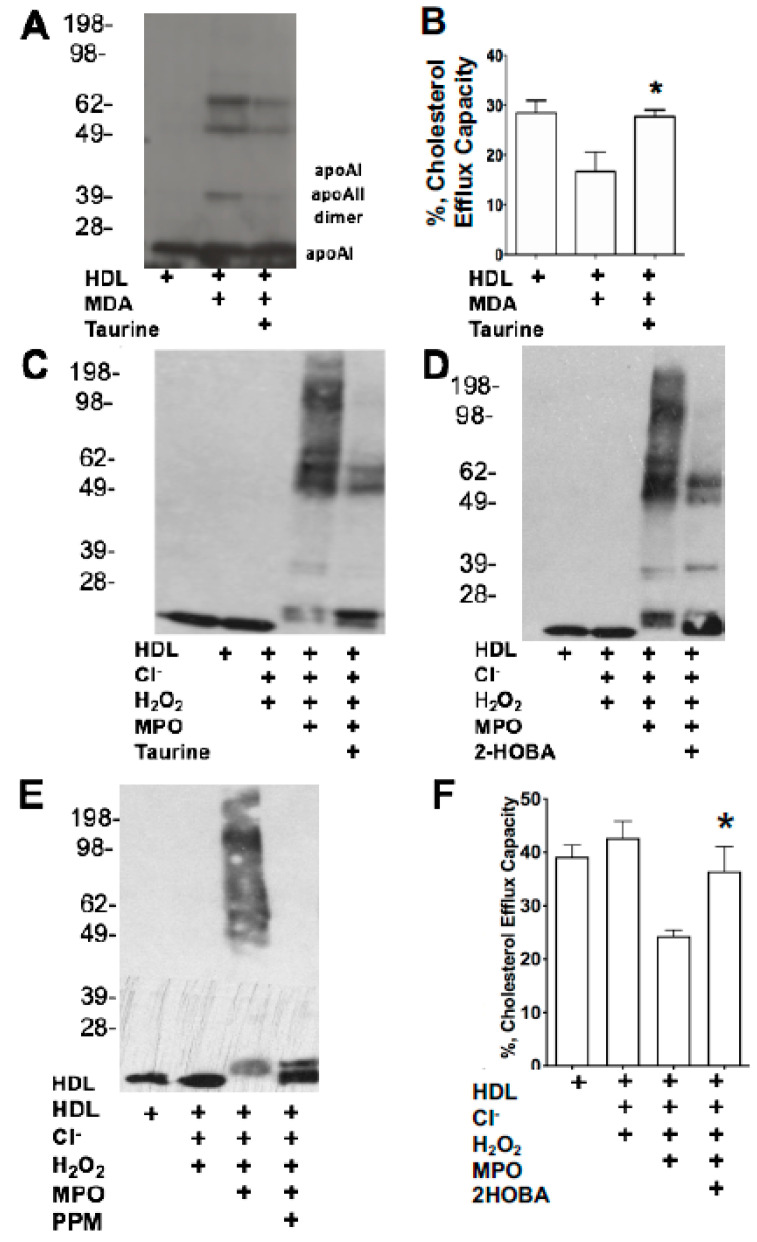
Effects of reactive dicarbonyl scavenging on HDL-apoAI crosslinking and cholesterol efflux capacity during MDA modification and MPO-mediated oxidation. ApoAI crosslinking (**A**) and cholesterol efflux capacity (**B**) were determined in HDL-treated with or without 5 mM taurine before the addition of MDA (250 µM MDA). The ability of reactive dicarbonyl scavenging with 5 mM taurine (**C**), 500 µM 2-HOBA (**D**), or 500 µM PPM (**E**) to prevent MPO-HDL-apoAI crosslinking was determined. The cholesterol efflux capacity of HDL modified with MPO in the absence and presence of 2-HOBA was measured in J774 cells (**F**). Graphs represent data (mean ± SEM) of the experiments; * *p* < 0.05 is indicated by comparing with MDA-HDL group (**B**) and with MPO-mediated HDL oxidation group (**F**) by one way ANOVA.

**Figure 4 nutrients-12-01937-f004:**
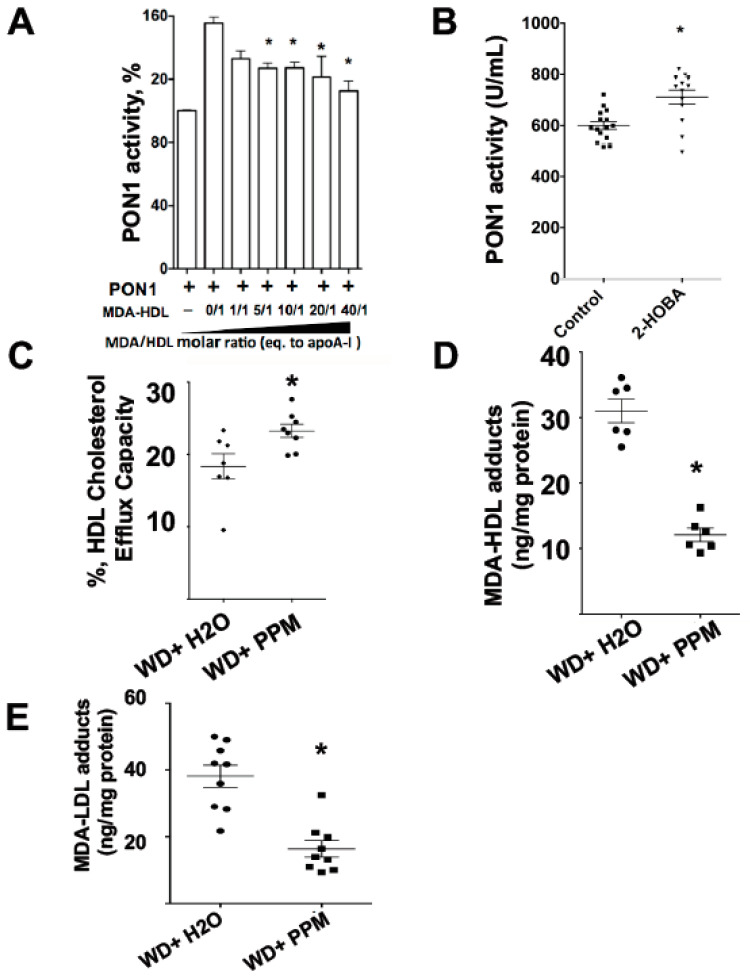
Effects of HDL MDA modification and reactive dicarbonyl scavenging on PON1 activity. The PON1 activity assay was performed using the EnzCheck PON1 activity kit (Thermo Fisher Scientific), which measures the organophosphatase activity of PON1. PON1 standards and diluted human serum samples were prepared in TSC buffer. The samples and standards were then incubated for 30 min with TSC buffer containing 100µM substrate. Fluorescence intensity was then measured using Biotek’s Synergy MX Microplate reader after 30 min of incubation at 37 °C. The effect of MDA modification of HDL on PON1 activity was measured (**A**). The PON1 activity was measured in plasma of *Ldlr^−/−^* mice (*n* = 15 in control, *n* = 13 in treatment group) consuming a western diet and treated with water alone or with the reactive scavenger, 2-HOBA (**B**). The cholesterol efflux capacity of HDL from *Ldlr^−/−^* mice treated with water alone or with the reactive dicarbonyl scavenger, PPM (**C**). MDA-HDL (**D**) and MDA-LDL (**E**) levels were measured in *Ldlr^−/−^* mice treated with the reactive dicarbonyl scavenger PPM using ELISA. Graphs represent data (mean ± SEM) of the experiments; * *p* < 0.05 is indicated by T-test or by comparing with HDL-PON1 group (**A**) one way ANOVA.

**Figure 5 nutrients-12-01937-f005:**
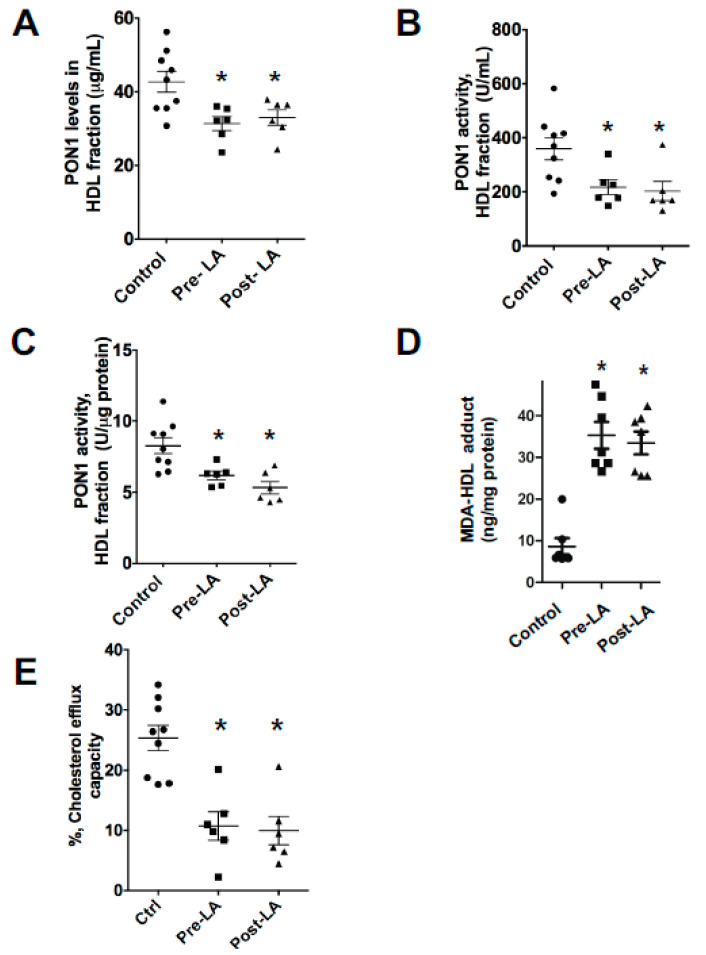
FH patients have decreased PON1 activity, increased MDA-apoAI adducts, and defective HDL cholesterol efflux capacity. The PON1 levels were determined in the ApoB-depleted serum fraction (**A**) from normal controls (*n* = 9) and FH (*n* = 6) subjects pre- and post-LDL apheresis. The PON1 activity was determined in ApoB-depleted serum fraction (**B**) and the specific activity of PON1 in control and FH subjects is shown (**C**). HDL-MDA content (**D**) and cholesterol efflux capacity (**E**) were determined in control versus FH subjects. The PON1 activity assay was performed using the EnzCheck PON1 activity kit, which measures the organophosphatase activity of PON1. PON1 standards and diluted human serum samples were prepared in TSC buffer. The samples and standards were then incubated for 30 min with TSC buffer containing 100 µM substrate. Fluorescence intensity was then measured using Biotek’s Synergy MX Microplate reader after 30 min of incubation at 37 °C. Graphs represent data (mean ± SEM) of the experiments; * *p* < 0.05 is indicated by comparison of control group with the indicated group using one way ANOVA.

**Figure 6 nutrients-12-01937-f006:**
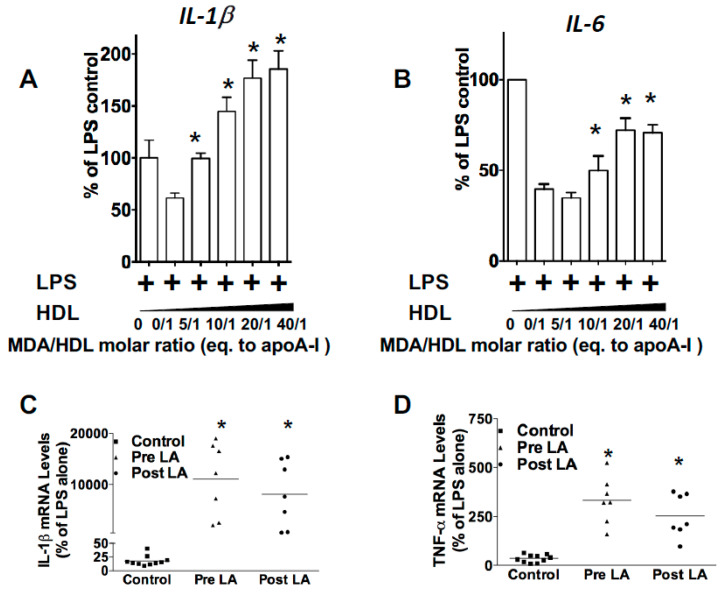
MDA-modified HDL increases expression of pro-inflammatory genes and FH-HDL exerts remarkable pro-inflammatory properties on *Apoe^−/−^* macrophages. HDL was modified with increasing doses of MDA and the effects on macrophage expression of IL-1β (**A**) and IL-6 (**B**) in response to LPS were measured. Graphs represent data (mean ± SEM) of the experiments; * *p* < 0.05 is indicated by comparing with HDL treatment group (**A**,**B**) using one way ANOVA. IL-1β (**C**) and TNF-α (**D**) mRNA levels were measured in *Apoe*^−/−^ macrophages treated with LPS and HDL from control (*n* = 9) and FH subjects (*n* = 7) pre- and post-LDL apheresis. * *p* < 0.05 is indicated by comparing with control group (**C**,**D**) using one way ANOVA.

**Figure 7 nutrients-12-01937-f007:**
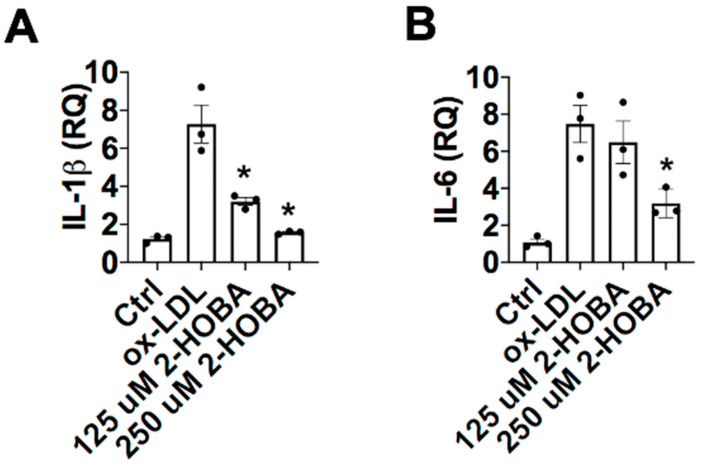
The mRNA levels of pro-inflammatory cytokine expression in *Apoe^−/−^* macrophages are significantly reduced by 2-HOBA. In the presence of 2-HOBA, macrophages were incubated with 50 μg/mL oxLDL for 24 h, and then the effects on macrophage expression of IL-1β (**A**) and IL-6 (**B**) in response to oxLDL were measured. Graphs represent data (mean ± SEM) of the experiments; * *p* < 0.05 is indicated by comparing with ox-LDL treatment group (**A**,**B**) using one way ANOVA.

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
