# Peer review of "Reactive Dicarbonyl Scavenging Effectively Reduces MPO-Mediated Oxidation of HDL and Restores PON1 Activity"

_nutrients, 2020, doi:10.3390/nu12071937_

Round 1

Reviewer 1 Report

In this work Huang et al. elaborated regulatory functions of PON1 and reactive dicarbonyl scavengers in HDL/ApoA-I oxidation pathway. The authors demonstrated that PON1 and reactive dicarbonyl scavengers abrogate the crosslink between ApoA-I and HDL or MDA-apoAI/HDL, rescue the cholesterol efflux, and as a consequence preserve the atheroprotective and anti-inflammatory bioactivity of HDL. Although the questions raised here are of importance in the context of hyperlipidaemia, however there is a lack of novelty in this study.   

Fig.2, 3, 4, the data is convincing. However, the results could be easily anticipated and are sort of something have been clarified previously. It is well established (which was also included in the ‘introduction’ part) that MPO oxidizes HDL and impairs cholesterol efflux. It’s also a fact that MDA binds to apoAI and serves a similar role. Only new finding is PON1 interrupts all these processes. The whole Fig.3, 4 can be moved to supplementary materials.  

Fig.6, what is the biological link between dicarbonyl scavenger and PON1? The authors showed that PPM played a protective role in Ldlr-/- model, by increasing HDL cholesterol efflux while reducing MAD-HDL adduct. Meanwhile, PON1 activity was upregulated in the present of 2-HOBA. But are the reactive dicarbonyl scavenger-mediated functional benefits dependent on PON1 activation? It would be helpful to introduce any form of PON1 knockdown (antibody, shRNA, gene depletion) in this model. Also, why the authors switched from 2-HOBA to PPM in the same model?

It’s a very interesting observation that PON1 level and activity decreased in FH patients. But again nothing associated with reactive dicarbonyl scavenger has been addressed. With only one animal model which is Fig.6 but no other data supporting, the current version is too preliminary to make a conclusion as what the title suggested.

Fig.8A,B, Did the authors try to add reactive dicarbonyl scavenger in this exp to see if it can block proinflammatory response of MDA-HDL to LPS?

The figure legends part should be reworded to add more information. Current version is very difficult to reference for accurate details.

Reviewer 2 Report

The present paper is a well presented body of work assessing the impact of MDA mediated damage on HDL/LDL and should be applauded on the combination of biochemical, molecular, in-vitro, in-vivo and clinical work. I recommend the work be published with minor revisions after consideration of the following points:

  1. The identity of the inhibitors should be stated in the abstract (i.e. 2-HOBA and PPM)
  2. More detail must be provided on how the animal studies were performed. How many animals were used in the study? Were any animals excluded from analysis? How were the animals sampled? How often were they sampled, or only at sacrifice? Why were there different numbers in the control and treatment groups?
  3. The scale in Fig 6A is obscured by the axis legend
  4. The authors should discuss in the text why they only studied PON1 activity in 2-HOBA treated mice, but studied a broader range of parameters for the PPM treated mice. Were these parameters tested in both experiments? If so, why were they not included? If not, why were they not performed? (e.g. sample limitations, cost, not relevant?)
  5. Typo in line 341
  6. Consider discussing the in-vivo relevance of MDA:HDL ratios. A 40:1 ratio clearly initiates very significant protein crosslinking. Has this been observed in animal or human studies? If so, this would increase the impact of the current work.
  7. Figures 5C and 5F reveal different efflux capacities for the HDL controls. Were these from separate experiments? Where does the variation come from? The intra-assay variation (error-bars) look very good, but there is a >10% difference between the two sets of data.
  8. The pattern of oxidation (oligomers, etc) on the gels from MPO-mediated oxidation vs. MDA treatment is quite different. This may indicate a difference in the underlying oxidation chemistry occurring under the MPO experiment in Fig2.
  9. The authors should consider commenting on the method of cross-linking under treatment with MPO. While the MDA treatments are likely lysine-mediated crosslinks, the MPO treatment may yield many alternatives via reaction with HOCl (e.g. di-tyrosine).

The role of hypochlorous acid, HOCl, is not adequately discussed as an alternative explanation for many of the results presented (especially for the in-vivo experiments). 2-HOBA and PPM are highly electron rich molecules which should be readily oxidised by HOCl. Any results observed in-vivo may also be due to the scavenging effect of 2-HOBA/PPM on HOCl, rather than MDA. Have the authors considered this point and do they have any supporting information which indicates a greater role of MDA over HOCl?

Reviewer 3 Report

Comments

  1. I find the topic, as a whole, interesting but I couldn’t understand the following points presented by the authors. So, minor changes should be done on discussion to assist readers understand the results correctly.

# MPO mediates the chlorination, nitration, and direct oxidation of proteins. Chlorination of apoA-I by the MPO pathway has been reported to impair its ability to remove excess cellular cholesterol by the ABCA1 pathway. It is also known that chlorination blocks the binding of apoA-I to ABCA1.

Figure 5 shows that the modification of apoA-I by MPO impairs the HDL-mediated cholesterol efflux. So, the authors support previous findings. However, this impairment was not observed in the presence of reactive dicarbonyl scavengers. The scavengers abolished not only MPO-mediated apoAI crosslinking but also MPO-mediated chlorination and nitration of apoAI?

# PON1 forms a ternary complex with apoAI and MPO. PON1 interaction with apoAI decreases MPO activity. Thus, HDL-associated PON1 is effective in preventing MPO-mediated HDL modification and apoAI crosslinking. On the other hand, MDA modification of HDL results in reduced ability to activate PON1. This means that PON-1 is effective in preventing MPO-mediated HDL modification if this modification level is low, but not effective if the modification progresses?

  1. Differences between mean values were determined by one-way ANOVA coupled with Bonferroni-Dunn post hoc (all the figures except Fig. 1).

# Is there any difference between * with under bar and *?

# What did authors compare to calculate P values (vs. what)? For example, Figure 2D.

  1. Please check English sentences again. For example,

# Line 103: Apoe-/- and Ldlr-/-? Ldlr-/- is LDL receptor knockout mice on a C57BL/6 background.

# Line 127: Ca2+ and Mg2+.

# Line 142: H2O2 should be added?

# Line 149: Delete “instructions”.

# Line 297: How to calculate 45% (p<0.05)?

# Line 341: Delete “in”.

# Line 376: from control (n=10) or pre (n=7) and post (n=7) LDL apheresis.
